# The Role of Artificial Intelligence in Personalizing Social Media Marketing Strategies for Enhanced Customer Experience

**DOI:** 10.3390/bs15050700

**Published:** 2025-05-19

**Authors:** Hasan Beyari, Tareq Hashem

**Affiliations:** 1Department of Administrative and Financial Sciences, Applied College, Umm al-Qura University, Makkah 24382, Saudi Arabia; 2Marketing Department, Faculty of Business, Applied Science Private University, Amman 11937, Jordan; t_hashim@asu.edu.jo

**Keywords:** artificial intelligence, social media marketing strategies, customer experience, customer purchase intention

## Abstract

This paper explores the role of artificial intelligence (AI) in personalizing social media marketing strategies and its impact on customer experience, with a focus on consumers within the MENA region. Using data collected from an online questionnaire completed by 893 individuals, the study confirms that AI significantly enhances social media marketing by offering personalized content, optimizing influencer selection, and enabling real-time consumer interaction. These capabilities not only increase customer awareness but also improve user experience and purchase intentions. Key AI tools such as influencer marketing, content optimization, and customization are effective in capturing consumer attention, although further research is necessary to deepen understanding. By examining AI’s ability to analyze vast datasets and support targeted marketing efforts, the study contributes to both academic and practical discourse, offering insights that businesses can use to refine their AI-driven social media strategies. Ultimately, the research aims to guide marketers through the complexities of AI deployment, ensuring its benefits are fully realized for consumers.

## 1. Introduction

The growing prominence and participation of social networking sites have gradually immensely impacted individual consumer buying behavior. The increasing integration of AI in marketing represents a new revolution in how firms engage customers, especially on social media. AI-powered tools may include influencer marketing, personalization, and content optimization to have the firm fine-tune its strategies by forecasting consumers’ behavior. This approach frames the content per their taste and preference, raising customer interaction quality. More than 80% of executives in the retail and consumer space are likely to leverage AI automation for business process improvement by 2025 ([4]). It has shown how fast the influence of AI took a great liking among marketing professionals to reduce time and improve operational work with a personalized customer experience. Even as the increase in AI adoption within marketing rises, significant flaws remain in the research that have not accounted for the various constituents of AI in terms of its impact on customers. While 88% of marketers claim that AI is used to manage the personalization of customer journeys across touchpoints, influencer marketing, personalization, and content optimization are separately under-explored in improving customer engagement ([36]). Companies also refer to the skill level and other privacy concerns as deterrents to adopting AI and point toward an investigation to further effectively implement the strategies.

In light of this view, this article deliberates on the role of AI in personalizing social media marketing strategy and what that would do to the customer experience. Antecedents of awareness, purchase intention, selection of the social media platform, and seeking information are mediators by which AI exerts an influence. The goal is to determine AI’s role in marketing and how it potently drives customer engagement and experience. This will contribute to the article’s theoretical and practical objectives and fill some gaps in the literature. The article contributes information that can be adopted by enterprises in efforts to improve their practices in social media marketing with AI. The researchers attempt to guide marketers through the intricacies of AI implementations so that this powerful technology can serve customers.

## 2. Literature Review

Several new marketing technologies modify the ways businesses interact with their customer base. Research by [13] ([13]) shows that AI personalization enhances engagement through preference-based content delivery, but [32] ([32]) explain how algorithms provide better influencer campaigns. [44] ([44]) demonstrate through recent analysis that algorithmic data processing leads to enhanced real-time decision-making, which improves promotional strategies for changing customer desire patterns. According to [6] ([6]), AI systems depend on organizations being ready to adopt them and their ability to attain skilled personnel who can operate these advanced systems. Hence, [24] ([24]) demonstrate that privacy, together with ethical concerns, needs resolution to build trust between consumers and AI-enhanced environments. Scientists have shown their dedication to upgrading traditional frameworks because of technological advancements and changing consumer behaviors.

### 2.1. AI in Marketing

AI is a new frontier in the marketing world where brands can build personalized and powerful strategies to help them connect with customers. It is a result of the capability of AI to gather and process big chunks of customer data while churning out highly targeted marketing strategies. Recommendations systems are an excellent example of the approach where artificial intelligence is employed to learn about customer behavior and, hence, offer products and content that would meet their needs. Considering the present market scenario, this is a monetarily feasible strategy wherein the expectation of the consumers for a personalized approach in offerings is relatively high. In their study, [24] ([24]) identified that AI improves customer engagement by mapping general consumer behavior, allowing marketers to offer a more apt and timelier proposition. Further, the application of AI in enriching customer journeys is evident from the fact that 88% of marketers use AI for the same purpose ([36]). AI shapes how customers are engaged and contributes operationally towards effectiveness. For instance, up to 54 percent of companies reduced their costs owing to the integration of AI ([19]). Therefore, AI seems to enhance marketing effectiveness. Real-time customer analytics enables AI to help companies fine-tune operations, change mid-stream marketing campaigns, and ramp up marketing efficiency ([44]). Thus, applications of Artificial Intelligence in marketing are consistently gaining importance in the market, with data-dominating brands that want to remain relevant in the present world.

### 2.2. Social Media Personalization

AI is best fitted for carving out social media marketing strategies in the present world. This helps the marketer churn out content that is apt and timely in consumers’ eyes. The offerings of products or services they need are provided to consumers at the right time. According to [40] ([40]), the likes, shares, and comments of every customer are analyzed with the help of AI algorithms to keep their needs in real-time ([13]). Up to 88% of marketers engaging with AI used it to achieve higher customer engagement through seamless brand experiences across many touchpoints ([36]). Integrating AI into influencer marketing can enable brands to identify the right influencer regarding audience demographics and engagement, making such campaigns even more effective ([32]). Additionally, it contributes to content enrichment because AI signals the types of posts that may receive high levels of engagement. This strategy allows for proper planning of social media and boosts returns.

### 2.3. Integration of AI in Marketing

The conventional marketing approach depends on general demographic classifications as well as basic promotional approaches that fail to serve diverse consumer groups effectively. The data-based approach from AI-driven marketing allows personalized message delivery through specific channels at proper moments, according to current data points ([13]). The advanced analytics system of AI influencer marketing allows it to find content creators who possess followers matching specific target demographics, according to [24] ([24]). This exceeds traditional celebrity endorsement reach limitations. The use of personalization through product recommendation technologies produces tailored suggestions that provide precise advertising strategies instead of using generic methods ([32]). [33] ([33]) explain how content optimization algorithms automatically modify marketing messages for maximum impact when users need specific information at a particular time and place. AI-based methods become more economically beneficial over time because machine learning provides cost-saving analytics that optimize trial costs and generates higher returns ([44]). The implementation of these artificial intelligence elements allows businesses to design marketing initiatives that connect and measure with their audience better than conventional methods, allowing them to establish AI-based marketing beyond traditional practices.

### 2.4. Challenges and Opportunities

Some barriers have always stood in the way of AI being an indispensable marketing tool. A significant challenge that organizations face is lacking knowledge of AI technology ([23]). As [19] ([19]) highlighted, 34% of businesses have reported limited AI skills and knowledge as the reason for their incorporating only a small extent in 2024. This implies that organizations cannot realize all AI benefits due to this skills gap. Most firms also cannot utilize AI’s sophisticated elements, which agrees with ([6]). But, a critical concern about using AI today is ethical issues related to data privacy. AI’s application in marketing raises the challenge of how data are used and managed, especially in those methods where companies collect and apply such data. In 2024, Statista reported that four in every ten marketers were concerned about data privacy and ethical issues linked to the use of AI in marketing ([36]). These concerns become greater because AI is used for mining and extracting information from large sets of personal data, a process that occurs mostly without express consent and usage transparency.

These issues, in the end, are challenges to the potential of innovation. The organization that trains the workforce in AI gets an appropriate and effective marketing plan by tapping into AI’s power in marketing. With rising ethical concerns regarding AI usage, practicing firms present a better relationship with consumers due to the explicit use of transparent consumer data, specifically because of rising reforms in data protection. As discussed in boardrooms, 8% of businesses will disrupt content creation and customer service ([28]). It will enable brands to offer experiences to people on a scale that was never possible before. Peer-reviewed studies confirm that AI marketing features comprising influencer marketing along with personalization and content optimization systems influence four main consumer results about brand recognition and buying intentions and platform choices together with information inquiry ([5]; [17]; [24]). The studied constructs derive their empirical basis from experimental research, but this paper uses their cohesive meaning in connection to the uses and gratification theory. The subsequent part of this study uses UGT to explain how AI instruments meet specific user needs while influencing marketing behavior decisions. This study aimed to discover how and where AI could be applied in marketing, considering its future implications.

## 3. Theoretical Background

### 3.1. Uses and Gratification Theory

The uses and gratifications theory (UGT) provides an essential theoretical understanding of how audiences purposefully choose media to achieve their wide range of requirements. The UGT theory was first introduced by [21] ([21]), which states that people use media to achieve goals that bring them gratification from escapist content while creating social relations and learning information. [1] ([1]) investigated consumer brand equity perceptions by bridging theories that related media usage with brand equity development. [18] ([18]) and [31] ([31]) conducted research that expanded the UGT framework through their inclusion of customer experience components, thus making it suitable for addressing current marketing obstacles. Research findings by [29] ([29]) and [35] ([35]) clearly explain how media consumption affects consumer choices regarding buying products. All these fundamental publications form an essential base for examining AI marketing strategies within modern digital marketplaces. Consumer interaction with AI-enhanced marketing is ideally analyzed through the UGT framework because this model recognizes users as self-directed seekers of media interactions that help them achieve goals ([1]). Through its application to AI features like personalization and influencer content, UGT permits researchers to comprehend what drives consumers to adopt particular actions by examining their behavior as strategic choices in pursuit of psychological, informational, and social needs.

### 3.2. Uses and Gratification Theory in Consumer Behavior

Uses and gratification theory (UGT) provides good insights into how the active consumer engages in media for certain purposes or to achieve certain goals. The theory derives from the communication science tradition and postulates that audiences do not receive media messages ([10]). Instead, consumers look out for media content to satisfy their psychological needs. Under the uses and gratification theory, individuals employ media technologies to meet their information, social, and hedonic requirements ([35]). The present age of AI-social media presents user needs as specific marketing features with which users can interact. Users require information-based content that satisfies their tastes to find useful value and improve their choices. Users’ social needs are fulfilled through influencer marketing since digital followers use platform creators to build relationships, gain social status, and find community values and ideal self-identity connections ([18]). Content optimization is crucial in hedonic needs because it delivers timely content that entertains viewers, creating visual appeal and intensive focus. AI tools make important connections that function as gratification agents, modifying user behavior while using platforms and assessing marketing decisions.

Mapping UGT on AI marketing components, such as influencer marketing, personalization, and content optimization, shows how such components are the leading drivers of customer engagement. AI influencer marketing, for example, stands for the UGT principle of social connection, which maintains influencer content that consumers try to establish themselves with influencers and other community members. AI personalization, the other key element, answers consumers’ calls for effective information searching and decision-making by matching their preferences with relevant content. As a result, this approach promises heightened product awareness and consequential purchase intention ([32]). AI also offers more platform options through content optimization techniques. In so doing, it modifies the format and time of exposure to best-fit patterns of user engagement, which is useful in presenting appealing content with suitable timing and place for the consumer ([24]). Relating these AI components to customer behavior, UGT is a theoretical framework through which AI can drive engagement and improve the general customer experience.

The AI components relate to the study mediators, which include awareness, purchase intention, platform choice, and information seeking to show how UGT influences consumer behavior. AI-designed personalization of brand messages leads to consumer awareness because this process satisfies their need for information ([11]). AI-driven targeting systems combine information-based and pleasure-based motivations to form customer purchase intentions, particularly when they show fascinating products that match individual preferences. Users choose platforms based on their desire for social engagement and hedonic satisfaction since they prefer platforms serving their community groups and offering well-curated content ([24]). Information seeking directly applies informational gratification because users depend on AI-driven recommendation engines and chatbots to obtain speedy, tailored responses requiring minimal cognitive resources. Through the UGT concept, researchers can develop a solid framework to explore and explain how these mediated behaviors evolve when AI-based marketing technologies exist.

### 3.3. UGT and Algorithmic Contexts

The process of algorithmic content selection within digital media changes the way users encounter and interact with digital content, specifically on social networking sites. The authors of [11] ([11]) argue that UGT focuses on audience-gratification-seeking activities even though AI systems utilize predictive analytics, which might present non-preferred content to users. The approach of algorithms that favors engagement metrics against user autonomy results in the restriction of actively selected content ([33]). [14] ([14]) mention that the algorithmic models strengthen echo-chamber effects, reducing the range of gratifications that users could have accessed. User-driven media consumption exists partially due to human agency but also results from data-driven automated recommendations. Researchers should examine the relationship between AI platforms and UGT theoretical framework since technology continues to advance.

### 3.4. UGT Justification for Study Variables

Each study mediator—customer awareness, purchase intention, social media platform selection, and information seeking—finds a foundation in UGT-based gratification types. Users depend on AI tools for informational contact by obtaining relevant brand content and product information through the awareness phase. Consumers’ purchase intention depends on hedonic and informational gratifications since AI technology sends personalized offers at opportune times ([24]). Consumers’ hedonic and social concept of information seeking exists entirely within informational gratification since AI components reduce search obstacles while delivering relevant content through chatbots and recommendations. The theoretical framework of UGT confirms that each mediator functions as an observed result in empirical studies and a theoretically correct customer choice behavior.

### 3.5. UGT Gratification Mapping

Table 1 demonstrates how the principal UGT gratification types unite with each AI element from this research. Aggravating user engagement with digital media depends on meeting their informational, social, and hedonic needs ([14]). [33] ([33]) and [35] ([35]) suggest that algorithmic personalization provides multiple gratifications for influencer marketing. The conceptual maps present unambiguous relations between the hypotheses and theoretical elements. The theoretical gratifications mentioned earlier serve as cornerstones for the study model to create hypotheses about how AI marketing features affect customer experiences. The table below presents the theoretical match between gratifications, AI aspects, and their corresponding study hypotheses. Therefore, this framework enhances the logical validity of the proposed conceptual framework.

## 4. Development of Hypotheses and Conceptual Model

The study forms its hypotheses by connecting each component of AI to consumer gratifications, which drive particular behavioral responses according to the UGT framework. The next section develops research hypotheses that explore how different AI components affect customer engagement through essential mediators after the gratification mapping and theoretical alignment concluding section.

### 4.1. AI Components in Customer Awareness

Customer awareness describes a potential consumer’s ability to remember the brand’s products or services. AI has increasingly become a tool of necessity in improving such awareness. This is mainly achieved using personalized content and targeted marketing. Various studies have indicated that AI plays a vital role in developing awareness, where specific information is provided to the consumer in a needed manner. In this regard, [12] ([12]) found that AI-based recommendation systems enhance brand visibility by recommending certain brands to users based on previous browsing and purchasing history. [26]’s ([26]) study also proved that AI algorithms, which empower branded content to reach select audiences, increase its reach and recognition in social media. Additionally, according to [3] ([3]), AI can analyze huge amounts of data in real-time. This enables marketers to deliver more accurate and relevant messaging, which is one of the key cases necessary for building brand awareness. Based on the findings, we hypothesize the following:

**H1.** 
*AI components, including personal recommendation and content optimization, positively influence customer awareness.*


### 4.2. AI Components on Customer Purchase Intention

Purchase intention refers to the probabilistic or chance-like measure of buying a specified product or service. Recommendation systems, personalized ads, and dynamic pricing models may influence purchase intention. AI-powered personal marketing enhances consumers’ purchase intention ([42]). Based on this type, recommendations are made to the user concerning their preference and online behavior. In [5]’s ([5]) study, AI-powered email marketing systems raise purchase intention. For [39] ([39]) also, AI-based dynamic pricing increases the probability of purchasing at the best price at the right time. These studies indicate that AI would enhance consumer purchase readiness because of time and objective interventions. Therefore, we would expect the following:

**H2.** 
*AI components, such as personalized recommendations and dynamic pricing, positively influence customers’ purchase intention.*


### 4.3. AI Components on Customer Social Media Platform Selection

Customer choice of social media channel refers to the process or procedure the customer undergoes in deciding what channel to use, depending on their preference and the content’s relevance. Most AI-driven algorithms, such as those used in optimization technologies of content, personalized recommendations, or targeted ads, have increasingly driven decisions built into the user experiences of the platforms ([33]). They create the most impact in choosing a particular platform since on some platforms—Instagram, Facebook, or X (formerly Twitter)—individual users are guaranteed to obtain information relevant to interests and habits. Equally, [25] ([25]) established that AI will most likely make personalized experiences across different platforms. This feature will make people stick with those offering more relevant content. AI can also interpret user behavior in real-time and enable marketers to create content on the platform that best engages them. This personalization instills specific platforms with greater value for the user, hence driving their selection ([24]). Based on these findings, we hypothesize the following:

**H3.** 
*AI components in the form of content optimization, along with personalized recommendations, influence customer social media platform selection.*


### 4.4. AI Components on Customer Information Seeking

AI components, such as chatbots, voice assistants, and recommendation systems, come with a place and gradually define how consumers search and retrieve this information. Customer information searching is the intentional process through which customers seek information to choose products or services ([12]). For example, [11] ([11]) established that AI-based chatbots improve efficiency in information search since customers obtain timely and precise responses through such engagements. [16] ([16]) further affirm that the AI recommendation system is crucially important in directing the consumer to the right information, as it gets to study their past search and browsing history. On the other hand, [9] ([9]) explain how voice search has made product information quite easy for consumers with voice assistants like Alexa and Siri because AI has made that process quite easy. These AI tools improve information accessibility and ensure that the knowledge provided to the user is accurate and relevant to their needs. Based on this literature, we would expect the following:

**H4.** 
*AI components, such as chatbots, voice assistants, and recommendation engines, positively influence customer information searches.*


### 4.5. Customer’s Awareness of Customer Experience

Customer awareness is the extent of knowledge customers possess about a brand or its product ([7]). The proportion of awareness forms an entry point at which a brand will engage an audience and develop perceptions and expectations of actions undertaken. It has also been proposed that the more engaging a customer is, the more likely they will attain positive brand interactions since the aware consumer is likelier to interact well with the brand ([27]). [30] ([30]) agree, arguing that awareness builds confidence and thus trust, hence improving the whole customer experience and satisfaction of the search and acquisition process for a product. Educated customers would also take it positively, according to [15] ([15]), because they are more aware of how to do it. Indeed, such a finding could regard customer awareness as one of the powerful tools that attracts potential customers while playing an important role in the development of good quality interaction of customers with a brand. Based on this, we postulate the following:

**H5.** 
*Customer awareness significantly and positively impacts customer experience.*


### 4.6. Customer Purchase Intention on Customer Experience

Customer purchase intention is the tendency of a consumer to buy any product or service directly associated with the general customer experience. Purchase intention is also the willingness of the consumer to buy the product, and it depends on the product’s attributes, such as usefulness, price, and promotion. According to [43] ([43]), positive purchase intention is related to a positive customer experience. Moreover, [38] ([38]) observed that with high purchase intention, the customer is more proactive and active. This evidence suggests that purchase intention does not mean just buying a product but also improving the relationship between the customers and the business and their satisfaction

Purchase intention functions as the theoretical basis for customer experience, but it exists before the customer experience phase. Purchase intention acts as the basis for this theory because it reveals positive pre-purchase evaluations, which function as essential elements prior to achieving superior customer experiences. The research conducted by [42] ([42]) shows that solid purchase intentions trigger both cognitive and affective processes that advance subsequent interaction engagement and satisfaction. Purchase intention functions as an initial agent that not only predicts buying actions but creates conditions that determine the quality of post-sale customer perceptions. Purchase intention also functions as an antecedent in our current conceptualization despite alternative research models where it is treated as a mediator. We therefore hypothesize the following:

**H6.** 
*Customer purchase intention positively influences customer experience, enhancing engagement, satisfaction, and transaction ease.*


### 4.7. Customer Social Media Platform Choice on Customer Experience

Choosing a social network means influencing the customer’s total experience because every network has its interface, format of content, and ways of engagement. Considering the examples above, the level of brand interaction varies among Instagram, Facebook, and X (formerly Twitter). The quality of the customer experience depends on it ([40]). The platform through which customers communicate goes a long way in determining how they view a company, as it defines either what type or what kind of communication the brand will have with the customer. [17] ([17]) established that buyers who engage with brands on platforms that align with their interests can ensure higher satisfaction and experiences. A perfect platform amplifies the quality of the communication channel to enrich the customer experience ([41]). Accordingly, we hypothesize the following:

**H7.** 
*Customer choice of social media platforms positively impacts customer experience.*


### 4.8. Customer Information Seeking on Customer Experience

Customer information search influences the consumer’s experience of a certain brand in the given market, as customers must seek information regarding the products they need. It would involve insight into the marketplace, which is important in introducing brand competition. Information search also ensures that the customers make the right choice so that the satisfaction with the brand is direct ([2]). More so, [22] ([22]) find that active information search improves purchasing decisions and the overall customer process. Hence, we hypothesize the following:

**H8.** 
*Information search on the part of customers positively influences customer experience.*


### 4.9. Conceptual Model

Figure 1 displays the conceptual model, which shows that AI influencer marketing, AI marketing personalization, and AI content optimization are connected to awareness and purchase intention, as well as social media platform choice and information seeking to affect customer experience. The research demonstrates that AI components affect mediators through the established relationships of H1 through H4. The mediators explained by H5 through H8 demonstrate their ability to mold the customer journey and produce effects on customer experience. The feedback mechanism in the model shows how user feedback shapes ongoing modifications of influencer campaigns, personalization techniques, and content optimization treatments. The model demonstrates how AI mechanisms adapt marketing practices through an ongoing cycle, which allows them to develop strategies according to shifting user requirements for more responsive customer paths.

## 5. Research Methodology

### 5.1. Research Design

The present study uses a quantitative approach; hence, the survey questionnaire examines the impact of AI components on customers’ experiences by moderating variables like awareness, purchase intention, and information seeking through social media platforms. Such a study can easily be subjected to a quantitative research design since the data could be accrued from many subjects and analyzed statistically to test hypotheses ([37]). These questions are closed-ended to assess the effectiveness of different AI components, influencer marketing, personalization, and content optimization on consumer behavior. The method is appropriate for understanding social media users’ perceptions, behavior, and attitudes in Saudi Arabia. Additionally, the study was ethically accepted by Umm Al-Qura University’s Deanship of Scientific Research Committee with informed permission. Furthermore, signed informed permission was acquired for participation in the research.

### 5.2. Sample Procedure, Data Collection, and Data Analysis Strategy

The research looked at people living in the MENA region. Researchers picked 1000 customers as a convenience sample and sent them an online survey through SurveyMonkey. The survey stayed open for six weeks. They analyzed 893 valid surveys, which made up 89.3% of the research group.

The survey had two main parts. The first part asked about things like age, gender, and education. The second part had statements about the research topics. People rated these statements using a 5-point Likert scale. The questionnaire’s final version had 33 items, as Table 2 shows.

The items of the survey are mentioned in Table 4. The management and analysis of primary data depended on the use of Statistical Package for Social Sciences (SPSS) v. 26 and the AMOS v. 23 program. Other statistical tests used in this study included frequency and percentages, mean and standard deviation, validity and reliability tests, and path analysis. The approval of construct validity depends on establishing precise operational definitions for each theoretical construct studied within the research. The research field benefits from exact conceptual boundaries, which enable the identification of overlapping dimensions. [33] ([33]) and [13] ([13]) highlighted that influencer marketing depends on trust factors combined with relevance elements and content credibility features. Two key elements that influence customer interaction exist within personalization, understanding individual choice preferences and content optimization, which dynamically improves message connection with use ([24]). The researchers [44] ([44]) stress that items must be clear to reliably capture constructs, while [6] ([6]) demonstrate their importance for accurate interpretation of data, particularly when working with AI in marketing. Each construct definition accompanies sample questionnaire items as well as corresponding reference details, which can be found in the table below. Research transparency increases significantly when detailed measurement elements are provided because these elements enable both replication and adaptation of research methods.

### 5.3. Data Analysis

The research used Principal Factor Analysis (PCA) to evaluate the measurement scales’ latent constructs while confirming their conceptual strength. The authors [37] ([37]) indicate that PCA works best at data reduction before applying Varimax rotation to enhance the interpretability of factor loadings. This research applied the Kaiser–Meyer–Olkin (KMO) measure to assess sampling adequacy and achieved verification of factorability through Bartlett’s test of sphericity, according to ([20]). [44] ([44]) state that multiple disciplines within marketing and social sciences adopt PCA with Varimax rotation because it produces factors that are both independent and straightforward to interpret. The research method follows the study direction to find separate constructs for AI marketing elements while maintaining simplicity in their interpretation. The applied methodology enhances the validity of the measurement model while producing dependable examination results.

## 6. Results

To achieve the study’s main goal, the researchers used frequency and percentages to break down the study demographics. Table 3 shows that men made up most of the respondents at 61.8%. Also, people aged 26–35 represented 49.8% of the sample. The largest group in terms of education had a bachelor’s degree, accounting for 42.6% of those surveyed.

### 6.1. Descriptive Statistics of Questionnaire

People surveyed have a positive outlook, as the mean and standard deviation indicate. A gap exists between the questions’ average and the scale’s set average. The study checked how reliable the measurement scales were for different variables using factor analysis results. The researchers used PCA extraction and Varimax rotation to obtain the most accurate data possible. Varimax rotation helps make sense of factor loadings by maximizing the spread of squared loadings across factors discussed by [34] ([34]) and a convergent validity test, also called a factor loading test, to check if the questionnaire was valid. Table 4 shows the test results. After careful thought, the researchers decided that items with loadings above 60% would count as real. They performed another test to check the scale’s reliability using composite reliability (CR), average variance extracted (AVE), and Cronbach’s Alpha. They used Cronbach’s Alpha to test the instrument’s reliability, obtaining a value higher than 0.70. This test proves that the instrument is reliable and fits the study. The CR test results show that the value is above the needed 0.70 mark ([8]).

The study conducted an analysis of discriminant validity to confirm that each construct is clearly differentiated from the other constructs ([8]). In order to conduct this analysis, the researchers assessed the correlation matrix of latent constructs. The diagonal elements of the matrix represent the square roots of the AVE. The correlations between constructs are displayed in the cells located outside the matrix’s lower left diagonal. This analysis emphasizes the importance of the shared variance between a construct and its measures being greater than the variance shared between the constructs and other constructs in the model. Therefore, discriminant validity is attained when the diagonal elements (square root AVE) surpass the off-diagonal elements in the corresponding row and column, as Table 5 illustrates.

The Bollen–Stine adjusted *p*-value was employed to evaluate the null hypothesis concerning the model’s correctness. The model exhibited a robust overall fit, evidenced by elevated values of GFI (0.981), NFI (0.922), TLI (0.971), CFI (0.936), AGFI (0.95), and an RMSEA value of 0.066. These values correspond with the suggested thresholds ([8]).

### 6.2. Hypotheses Testing

The statistical significance of the SEM’s structural loads was evaluated to measure its performance. Table 6 presents the results of the study hypotheses and the structural equation analysis that was conducted. In evaluating relationships, it is crucial to assess the *p*-value for each variable. A *p*-value below 0.05 signifies the presence of significant correlations. Each relationship in this study is significant.

The literature and statistical results confirm that Hypothesis 1, which suggests that AI components, including personal recommendation and content optimization, positively influence customer awareness, is supported (direct effect = 0.782; *p*-value ≤ 0.001). The results obtained in this study supported Hypothesis 2, which suggests that AI components, such as personalized recommendations and dynamic pricing, positively influence customers’ purchase intention. This finding is consistent with previous research and is further supported by the statistical results (direct effect = 0.869; *p*-value ≤ 0.001), which aligns with other recent studies in the field. The study findings strongly support the idea that AI components in the form of content optimization along with personalized recommendations influence customer social media platform selection, as previously indicated in the literature. The results of this study (direct effect = 0.83; *p*-value ≤ 0.001) provide strong evidence for this relationship. In addition, the obtained results confirmed that Hypothesis 4 holds true, indicating that AI components, such as chatbots, voice assistants, and recommendation engines, positively influence customer information searches. The results of Hypothesis 4 (direct effect = 0.572; *p*-value ≤ 0.001) are consistent with recent studies.

The literature and statistical results confirm that customer awareness significantly and positively impacts customer experience. The direct effect is 0.077, with a *p*-value of ≤0.001. The results of Hypothesis 6 indicate that customer purchase intention positively influences customer experience, enhancing engagement, satisfaction, and transaction ease (direct effect = 0.067; *p*-value ≤ 0.001), which aligns with previous research findings. In addition, Hypothesis 7 suggests that customer choice of social media platforms positively impacts customer experience. This was confirmed by the statistical findings (indirect effect = 0.066; *p*-value ≤ 0.001) and aligns with previous studies. The results obtained confirm that information search on the part of customers positively influences customer experience, supporting Hypothesis 8. This finding is consistent with recent studies in the field (direct effect = 0.675; *p*-value ≤ 0.001). The results reached are summarized in Figure 2 above, based on the presented tests of hypotheses.

## 7. Discussion

AI significantly impacts social media marketing by customizing methods to boost customer experience. By 2025, more than 80% of executives in retail and consumer industries plan to use AI to improve how their businesses work.

AI tools, like influencer marketing, personalization, and content optimization, have not been sufficiently studied but play a key role in motivating customer involvement. This article looks at how AI customizes social media marketing and how this affects customer experience. AI gathers and studies substantial amounts of data to create marketing plans that fit each customer, attracting more people. Eighty-eight percent of marketers use AI to improve customer journeys and connect with customers at different points. Real-time information from AI helps companies to run better and market more. When AI personalizes social media, it helps share relevant content with customers. AI systems look at likes, shares, and comments to understand what customers need immediately. AI influences influencer marketing by helping marketers identify suitable influencers based on audience data. Despite its benefits, however, many companies struggle to adopt AI due to a lack of tech know-how. Privacy concerns pose a significant challenge when using AI to achieve marketing goals. Companies that train their staff in AI can create effective marketing plans and build stronger customer relationships. UGT explains why people use AI products to meet specific needs or goals. Using AI in marketing raises questions about data use transparency and obtaining proper consent. Companies face hurdles because they lack the skills and knowledge to use advanced AI tools.

AI’s ability to obtain personal information without asking makes people more concerned about data privacy. Teaching workers about AI can lead to new marketing strategies that make customers happier.

Components of AI and Customer Awareness

AI-powered influencer marketing, marketing personalization, and AI content optimization influence customer awareness. AI looks at customer data to suggest products and services that match people’s likes and habits. Personalization helps customers discover new products and increases engagement and conversion. AI systems also improve content delivery based on user behavior and preferences. Customers receive the right information when they need it, which helps them understand their options and recognize brands better. Companies can also engage customers more through AI-powered personalized marketing. Emails, alerts, and ads customized for each person help to introduce products and services. AI tools examine large datasets to identify patterns in customer behavior. These insights allow businesses to adjust their marketing strategies to reach their target audience and make their brand more recognizable. AI-powered features like responsive chatbots or customized browsing experiences encourage customers to learn more about the brand and its offerings.

AI Components and Customer Purchase Intention

AI influences the study of huge consumer datasets to spot patterns and give researchers a better grasp of how customers act. This idea covers tracking online actions, what people buy, and how much they engage. AI tools can check customer feelings across their messages (like social media and emails) to gauge interest and intent. By looking at language and tone, businesses can spot potential buyers. AI also helps create custom marketing plans by suggesting products based on each consumer’s actions and likes. This personal touch makes people more likely to buy, as they feel closer to what is on offer. AI uses past data to predict future buying habits. By looking at trends and what consumers do, companies can guess when a customer is most likely to buy and tweak their marketing efforts. AI also spots key moments in a consumer’s journey that show they are ready to buy (like searching for product info). Spotting these small moments lets marketers reach out to customers at crucial decision points. AI sorts customer messages into leads, questions, complaints, or praise. This sorting helps to focus on urgent replies, and high-intent leads are more likely to turn into sales. By using AI-powered insights, companies can be more open about how they work (like return policies and data privacy), building trust with consumers—a key factor that shapes buying intent.

AI Components and Customer Social Media Platform Selection

When choosing a social media platform, companies should consider several AI features that can boost their marketing strategies and conversations with customers. AI tools impact analyzing user information to identify target groups, what they like, and how they act. This helps businesses choose platforms where their possible buyers spend time the most.

AI tools help marketers produce tailored posts for their audience. Content creation platforms with AI capabilities can enhance the creative process. AI monitors engagement across platforms, indicating which channels are most effective in reaching and involving people. AI sentiment analysis enables brands to grasp how people feel about their products or campaigns on various social media sites, leading to smarter strategy decisions. Choosing platforms with advanced AI automation can increase productivity by simplifying post-scheduling and customer question responses. AI analytics assess ad cost-effectiveness across platforms and guide companies to invest in channels with the highest return on investment. Clever AI algorithms can identify upcoming trends from current data, allowing marketers to stay ahead by selecting platforms that match future consumer needs. By incorporating these AI elements into their selection process, businesses can make wise choices about which social media platforms will best support their marketing goals and increase customer engagement.

AI Components and Customer Information Seeking.

AI tools, such as machine learning, have an influence on personalized recommendations by using customer data. Personalization helps users access information by offering relevant content based on their likes and previous actions. Natural language processing enables AI to grasp customer questions. This allows customers to ask questions to AI-driven systems, like chatbots and digital assistants, boosting their experience and satisfaction.

AI can spot patterns and trends in large datasets to figure out customer needs. Businesses can predict customer needs by using this data. AI-powered tech lets customers ask for help at any time. Today’s customers want quick answers, so round-the-clock access meets their expectations.

AI systems get better at answering questions by learning from their chats with clients. This ongoing learning helps them give better answers in the future. This makes clients feel more satisfied when they are looking things up. AI helps customers find more information. It customizes experiences based on how people talk, looks at data to find useful products, is always on, and gets better as people use it and give feedback.

Customer Awareness and Customer Experience.

Commercials, social platforms, and casual chats often shape a customer’s first thoughts about a business. These initial views color their feelings later on. People usually prefer a brand when its message lines up with what they experience. Also, people who know a brand think about it more. If later encounters fall short of what the ads promised, buyers might feel cheated. Marketing claims should match the real quality of goods or services. Good experiences lift a brand’s image as people talk it up with friends and share it online. Satisfied customers often discuss the brand in person and on the web, which helps spread the news. But, if many unhappy clients voice their problems, it can hurt customer loyalty and brand awareness. Customer support also affects both recognition and experience. Good interactions with brands create eager fans who then boost awareness and trust among possible buyers who value tips from others.

Customer Purchase Intention and Customer Experience

Customer experience has a big impact on buying intention, but it is not simple. When people enjoy their interactions with a brand, they are more likely to want to buy. This can happen through good marketing, easy-to-use websites, or great customer service. These good experiences make people think well of the company. As a result, they feel a stronger urge to make a purchase.

The emotional ties formed during customer interactions have a strong influence on purchasing choices. Companies that create emotional links with customers can build loyalty and push them to buy again. For example, when customers feel valued and understood in their dealings with a business, they tend to choose that brand over others when they need to buy something. People judge their experiences by comparing what they expect to what happens. When customers feel a company meets their needs, like receiving good products or helpful service, this match makes them surer about buying from that company later on.

Customer Social Media Platform Choice and Customer Experience.

The connection between a person’s social media platform choice and their experience is complex and diverse. This relationship is influenced by user preferences, brand engagement strategies, and specific features of various platforms.

In summary, many elements shape how a customer’s social media platform selection impacts their experience. It hinges on companies’ understanding of their audience’s needs, their capacity to tweak strategies, their reaction to interactions, their efforts across channels, their application of feedback to improve, their ability to create communities around products, and their talent for building trust through honest, open dialogues.

Customer Information Seeking and Customer Experience.

The connection between how customers research and their overall experience is not straightforward. When shoppers look into product details before buying, they pick items that match what they want and expect. This careful way of shopping has an impact on how happy customers are, as people feel sure about what they have chosen. Good information gathering helps buyers set real expectations for products or services. Businesses that give clear, easy-to-find information that fits these expectations make the whole customer trip better.

Shoppers who do their homework often start to trust brands that are open about what they sell. Trust is a critical art of keeping customers happy; when people feel they can count on a brand to give honest information, they are more likely to buy from them again.

Information-seeking creates a feedback loop where customers tell others about their experiences through reviews or word-of-mouth recommendations. Good experiences result in positive reviews, which push potential customers to look for more details.

Companies that know how customers search for information can shape their marketing strategies. By looking at data on common questions or concerns during the search stage, businesses can improve their communication and the overall customer experience.

## 8. Conclusions, Implications, and Limitations

AI tech, such as influencer marketing, personalization, and content optimization, catches clients’ eyes, but it needs more research. This article examined how AI shapes social media marketing and customer experience. AI processes huge amounts of data to build customer-specific marketing plans that attract more people. AI assists marketers in streamlining customer journeys and reaching out to customers at different stages, and real-time AI data enhances operations and marketing. AI tweaks social media to serve up content that matters to customers, and likes, shares, and comments allow AI systems to identify what clients want. AI aids advertisers in finding influencers who resonate with their audience. Yet many companies do not adopt AI because they lack tech skills, despite its benefits, so they can struggle because they lack the advanced AI skills they need. While privacy concerns make AI marketing challenging and AI in marketing raises concerns about transparency in data use and obtaining consent, staff who understand AI can create effective marketing plans and forge stronger client bonds. Ultimately, people use AI products to meet specific needs or goals that are in line with the concepts of use and gratification.

The uses and gratifications theory (UGT) claims that viewers seek out media to meet specific needs such as getting information, having fun, or connecting with others. This view assumes that users have much control over their choices. However, when AI systems decide for users—through recommendation algorithms, personalized search results, or content curation—this control may become unclear.

The study offers a tried-and-true paradigm that connects AI features like content optimization and personalization to the customer experience through factors like awareness and intent to buy. This deepens our comprehension of the ways in which AI impacts online marketing and consumer habits.

UGT’s concept of active audience choice interacts with AI-driven decision-making by utilizing less freedom due to algorithmic filtering, personalization vs. filter bubbles, hidden rewards via AI suggestions, shared control and collaboration, clarity, and perceived autonomy.

To implement this, marketers should focus on AI tools like custom recommendations and flexible pricing to boost customer awareness and their urge to buy. Using chatbots for quick chats can help improve how people look for information.

The limitation of this study is that it looked at the MENA region. New studies could check how AI adoption and customer behavior differ across cultures. In addition, relying on survey answers might lead to biases, like people trying to look good. Long-term or test-based studies could prove cause-and-effect links. Moreover, the research looked at three AI tools: influencer marketing, personalization, and content improvement. Future work could explore newer tech like AI that creates content or voice helpers. While the study mentioned privacy issues, we need to take a closer look at the ethical boundaries of AI (like unfair algorithms or keeping data safe). Likewise, AI and social media platforms are advanced and require ongoing research to keep up with tech progress.

## Figures and Tables

**Figure 1 behavsci-15-00700-f001:**
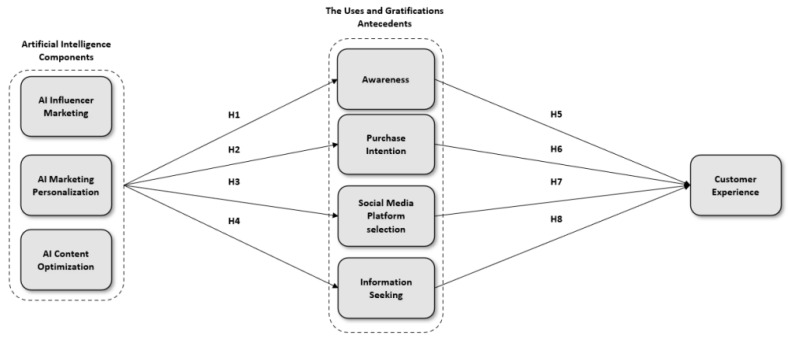
Conceptual model of study.

**Figure 2 behavsci-15-00700-f002:**
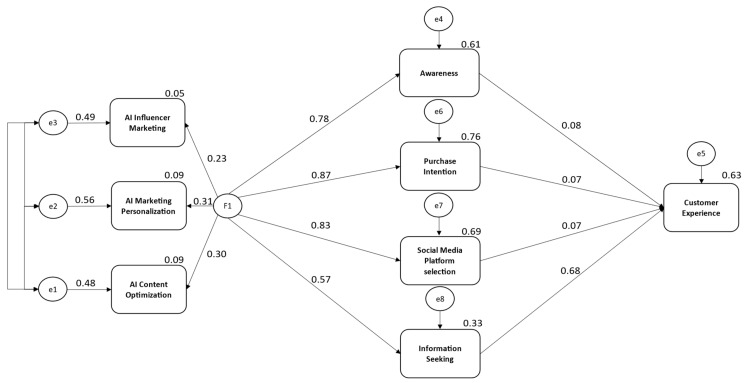
Path analysis.

**Table 1 behavsci-15-00700-t001:** Ways the principal UGT gratification types.

Gratification Type	AI Component(s)	Hypotheses	Rationale
Informational	Personalization, Content Optimization	H1, H4	Users demand timely and accurate products along with brand information to fulfill their requirements. Personalization techniques customize communication messages, and content optimization ensures that all material remains suitable for recipients.
Social	Influencer Marketing	H2, H3	Users want bonding experiences and a sense of community, which is represented by this factor. The use of influencer marketing depends on social connections to affect both purchase preferences (H2) and the selection of online platforms (H3).
Hedonic	Influencer Marketing, Content Optimization	Potentially H2 or H3	Captures the user’s pursuit of enjoyment and entertainment. Malfunctioning AI-driven influencer advertising, in combination with dynamically generated content, creates pleasurable experiences that lead customers to greater interactions and opt for specific choices.

**Table 2 behavsci-15-00700-t002:** Questionnaire items according to controls.

Variable	# of Statements
AI Influencer Marketing	5
Marketing Personalization	4
AI Content Optimization	5
Awareness	3
Purchase Intention	5
Media Platform Selection	3
Information Seeking	4
Customer Experience	4
Total	33

**Table 3 behavsci-15-00700-t003:** Descriptive of demographics.

	F	%
Gender		
Male	552	61.8
Female	341	38.2
Age		
18–25 years	69	7.7
26–35 years	444	49.8
36–45 years	234	26.2
above 45 years	146	16.3
Qualification		
High school or less	47	5.3
Diploma	94	10.5
Bachelor	380	42.6
Master	235	26.3
PhD	137	15.3

Source: SPSS v26 output.

**Table 4 behavsci-15-00700-t004:** Questionnaire analysis.

	Items	Μ	σ	FL	KMO Test	AVE	(CR)	α
IM1	Influencers driven by AI raise my awareness regarding products and companies.	3.476	1.071	0.899	0.858	0.743	0.935	0.913
IM2	I depend on the product advice provided by influencers driven by AI.	3.244	1.023	0.864				
IM3	Content produced by AI-driven influencers strikes a chord with me.	3.289	1.113	0.838				
IM4	AI influencers deliver useful details on the products I require.	3.286	1.089	0.892				
IM5	AI influencers motivate me to interact with brands via social platforms.	3.447	0.994	0.816				
AI Influencer Marketing (IM)		3.348	0.912					
MP1	Advertisements tailored by AI are very aligned with what I like.	3.550	1.011	0.790	0.631	0.677	0.893	0.838
MP2	I develop a stronger bond with brands that apply AI for personalized content.	3.568	0.996	0.845				
MP3	Using AI personalization simplifies the process of uncovering the items I am looking for.	3.570	1.176	0.809				
MP4	The use of AI boosts my experiences while shopping on social media.	3.563	1.156	0.845				
Marketing Personalization (MP)		3.563	0.892					
CO 1	AI-driven content engages users better than conventional content.	3.623	1.082	0.867	0.771	0.728	0.930	0.902
CO 2	I choose companies that apply AI for better content personalization.	3.564	1.063	0.883				
CO 3	I receive the most appropriate content from AI at the correct moment.	3.592	1.001	0.811				
CO 4	AI shares material that matches my choices on social platforms.	3.484	1.136	0.789				
CO 5	The content that AI enhances is thrilling for me to browse.	3.763	0.888	0.911				
AI Content Optimization (CO)		3.608	0.879					
A1	AI marketing has raised my knowledge of various brands.	3.980	0.889	0.898	0.74	0.797	0.922	0.872
A2	I recognize the items I truly need owing to AI-generated advertisements.	3.843	0.851	0.882				
A3	I find out about fresh products and services daily thanks to AI marketing.	3.746	0.910	0.898				
Awareness (A)		3.856	0.789					
PI 1	I tend to purchase items endorsed by AI technology.	3.470	0.910	0.824	0.771	0.727	0.93	0.903
PI 2	Advertisements powered by AI persuade me to buy items.	3.581	0.923	0.859				
PI 3	My confidence in deciding what to purchase rises when AI guides me.	3.719	0.835	0.939				
PI 4	AI revealed items to me that I ultimately decided to purchase.	3.583	0.916	0.835				
PI 5	I consistently respond to advertisements from AI.	3.372	0.956	0.800				
Purchase Intention (PI)		3.546	0.772					
MPS 1	I choose social media platforms that utilize AI to deliver important content.	3.648	0.864	0.912	0.742	0.69	0.916	0.742
MPS 2	Content created by AI shapes how I select my social media platforms.	3.713	0.859	0.908				
MPS 3	I frequently engage with brands on platforms operating with effective AI.	3.471	0.901	0.834				
Media Platform Selection (MPS)	AI revealed items to me that I ultimately decided to purchase.	3.611	0.775					
IS 1	I have better access to the information I want about products thanks to AI tools.	3.872	0.888	0.859	0.771		0.930	0.902
IS 2	I depend on chatbots and recommendation tools for product details.	3.497	0.929	0.843				
IS 3	With AI, I can discover more accurate and meaningful information about products.	3.918	0.833	0.760				
IS 4	AI systems enhance the way we access data across social media platforms.	3.882	0.867	0.875				
Information Seeking (IS)		3.793	0.735					
CE 1	My overall interaction with brands is improved by AI.	3.666	0.911	0.862	0.79		0.902	0.79
CE 2	Thanks to AI personalization features, my encounters with brands have become better.	3.610	0.928	0.881				
CE 3	AI improves my engagement with brands and makes it easy for me.	3.787	0.773	0.858				
CE 4	I experience increased happiness with firms that utilize AI to enrich my experience.	3.709	0.686	0.738				
Customer Experience (CE)		3.693	0.693					

Source: SPSS v26 output. Notes: A = Cronbach’s Alpha. FL = Factor Loading. AVE = Average variance extracted. (CR) = Composite Reliability.

**Table 5 behavsci-15-00700-t005:** Discriminant validity.

	AI Influencer Marketing	Marketing Personalization	AI Content Optimization	Awareness	Purchase Intention	Media Platform Selection	Information Seeking	Customer Experience
AI Influencer Marketing	0.862							
Marketing Personalization	0.517 **	0.823						
AI Content Optimization	0.585 **	0.485 **	0.853					
Awareness	0.193 **	0.209 **	0.219 **	0.893				
Purchase Intention	0.189 **	0.223 **	0.270 **	0.671 **	0.853			
Media Platform Selection	0.198 **	0.268 **	0.210 **	0.653 **	0.702 **	0.885		
Information Seeking	0.133 **	0.272 **	0.243 **	0.346 **	0.496 **	0.453 **	0.83	
Customer Experience	0.153 **	0.257 **	0.246 **	0.399 **	0.504 **	0.471 **	0.783 **	0.837

** Correlation is significant at the 0.01 level (2-tailed).

**Table 6 behavsci-15-00700-t006:** Hypothesized relationships.

			Standardized Direct Effect	SE	CR	*p*	Result
Awareness	<---	AI components	0.782	0.277	8.413	***	Supported
Purchase Intention	<---	AI components	0.869	0.297	8.600	***	Supported
Media Platform Selection	<---	AI components	0.83	0.290	8.412	***	Supported
Information Seeking	<---	AI components	0.572	0.188	8.214	***	Supported
Customer Experience	<---	Information Seeking	0.675	0.024	26.592	***	Supported
Customer Experience	<---	Media Platform Selection	0.058	0.028	2.074	0.038	Supported
Customer Experience	<---	Purchase Intention	0.06	0.029	2.023	0.043	Supported
Customer Experience	<---	Awareness	0.067	0.027	2.520	0.012	Supported

The support/not support significance reflected by *p* value *** *p* < 0.001.

## Data Availability

Data presented in this study are available upon request from the corresponding author.

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
