# Peer review of "The Role of Artificial Intelligence in Personalizing Social Media Marketing Strategies for Enhanced Customer Experience"

_behavsci, 2025, doi:10.3390/bs15050700_

Round 1

Reviewer 1 Report

Comments and Suggestions for Authors

Please see the pdf file

Author Response

Dear Reviewer 1,

Thanks for your valuable comments. Please see attached file.

Regards,

Reviewer 2 Report

Comments and Suggestions for Authors
  • I appreciate the opportunity to review the article titled “The Role of Artificial Intelligence in Personalizing Social Media Marketing Strategies for Enhanced Customer Experience”. The authors look to develop a framework for personalizing social media marketing strategies with artificial intelligence. Surveys are used to examine how artificial intelligence is integrated into constructs related to marketing to affect constructs from Use and Gratifications Theory and customer experience.
  • Strengths: The subject of study is highly relevant at this time. The survey method and SEM analysis are appropriate.
  • Major Weaknesses:
    • The theoretical foundation is lacking in its depth of literature review. Specifically, the seminal articles for the UGT theory, the core theory for this paper, are not referenced. In addition, articles related to customer experience, awareness, and purchase intentions are missing.
      • Katz, Elihu, and David Foulkes. "On the use of the mass media as “escape”: Clarification of a concept." Public opinion quarterly3 (1962): 377-388.
      • Aaker, David A. "The value of brand equity." Journal of business strategy4 (1992): 27-32.
      • Homburg, Christian, Danijel Jozić, and Christina Kuehnl. "Customer experience management: toward implementing an evolving marketing concept." Journal of the Academy of Marketing Science45 (2017): 377-401.
      • Palmer, Adrian. "Customer experience management: a critical review of an emerging idea." Journal of Services marketing3 (2010): 196-208.
      • Morrison, Donald G. "Purchase intentions and purchase behavior." Journal of marketing2 (1979): 65-74.
      • (Purchase survey reference) - Spears, Nancy, and Surendra N. Singh. "Measuring attitude toward the brand and purchase intentions." Journal of current issues & research in advertising2 (2004): 53-66.
    • The research methodology leaves several questions related to the study. First, who are the customers that were picked? The authors mention that 1,000 surveys were sent to customers, but they do not include where the customers are selected. The statements about the research products are not included. What statements were included? How was the validity of the statements established? How were the statements classified to the specific variables?
  • Minor Weaknesses:
    • For the research model, why is the purchase intention included as an antecedent to the customer experience?
    • Please use consistent language when referencing the survey/questionnaire. This is not a big issue, but can help with the readability.
    • Why was a level of 0.40 selected for the Principal Component Analysis (PCA) loading? The norm in this case tends to be 0.60.
  • Overall, I cannot support this article moving forward in the publication process. I would encourage the authors to narrow their focus on which component of marketing would be integrated with artificial intelligence and clearly delineate the difference between previous marketing literature and the contribution that will come from artificial intelligence. In addition, a full literature review could lend a foundation on which to develop the intended theoretical contribution. I have included a number of references, but the list is by no means exhaustive. The methodology needs to have additional details on the survey items and participant selection. I wish the authors the best of luck moving forward, a narrowed focus with a more in-depth examination of UGT and previous marketing literature can hopefully help to develop the article for a future submission.

Author Response

Dear Reviewer 2,

Thanks for your valuable comments. Please see attached file.

Regards,

Round 2

Reviewer 1 Report

Comments and Suggestions for Authors

No additional comments. Congrats.

Author Response

Dear Reviewer 1,

Thanks for the valuable comments. 

Regards,